# Role of the Redox State of Human Peroxiredoxin-5 on Its TLR4-Activating DAMP Function

**DOI:** 10.3390/antiox10121902

**Published:** 2021-11-27

**Authors:** Mégane A. Poncin, Pierre Van Meerbeeck, Joshua D. Simpson, André Clippe, François Tyckaert, Fabrice Bouillenne, Hervé Degand, André Matagne, Pierre Morsomme, Bernard Knoops, David Alsteens

**Affiliations:** 1Louvain Institue of Biomolecular Science and Technology, Université catholique de Louvain, 1348 Louvain-la-Neuve, Belgium; megane.poncin@uclouvain.be (M.A.P.); pierre.vanmeerbeeck@uclouvain.be (P.V.M.); joshua.simpson@uclouvain.be (J.D.S.); andre.clippe@uclouvain.be (A.C.); francois.tyckaert@uclouvain.be (F.T.); herve.degand@uclouvain.be (H.D.); pierre.morsomme@uclouvain.be (P.M.); 2Centre for Protein Engineering, InBioS, University of Liëge, Building B6C, Quartier Agora, Allée du 6 Août, 13, 4000 Liëge (Sart-Tilman), Belgium; f.bouillenne@uliege.be (F.B.); amatagne@uliege.be (A.M.)

**Keywords:** peroxiredoxins, PRDX5, Toll-like receptor, TLR4, DAMP, redox, cysteine residue, atomic force microscopy, SMFS

## Abstract

Human peroxiredoxin-5 (PRDX5) is a unique redox-sensitive protein that plays a dual role in brain ischemia-reperfusion injury. While intracellular PRDX5 has been reported to act as a neuroprotective antioxidative enzyme by scavenging peroxides, once released extracellularly from necrotic brain cells, the protein aggravates neural cell death by inducing expression of proinflammatory cytokines in macrophages through activation of Toll-like receptor (TLR) 2 (TLR2) and 4 (TLR4). Although recent evidence showed that PRDX5 was able to interact directly with TLR4, little is known regarding the role of the cysteine redox state of PRDX5 on its DAMP function. To gain insights into the role of PRDX5 redox-active cysteine residues in the TLR4-dependent proinflammatory activity of the protein, we used a recombinant human PRDX5 in the disulfide (oxidized) form and a mutant version lacking the peroxidatic cysteine, as well as chemically reduced and hyperoxidized PRDX5 proteins. We first analyzed the oxidation state and oligomerization profile by Western blot, mass spectrometry, and SEC-MALS. Using ELISA, we demonstrate that the disulfide bridge between the enzymatic cysteines is required to allow improved TLR4-dependent IL-8 secretion. Moreover, single-molecule force spectroscopy experiments revealed that TLR4 alone is not sufficient to discriminate the different PRDX5 redox forms. Finally, flow cytometry binding assays show that disulfide PRDX5 has a higher propensity to bind to the surface of living TLR4-expressing cells than the mutant protein. Taken together, these results demonstrate the importance of the redox state of PRDX5 cysteine residues on TLR4-induced inflammation.

## 1. Introduction

Peroxiredoxins (PRDXs) constitute a large superfamily of highly diverse and multifunctional peroxidases, expressed as six isoforms in mammals (PRDX1–6). Their intracellular functions are related to the redox state of their catalytic cysteine residues (Cys), of which there are one in PRDX6 and two in PRDX1–5. Intracellularly, PRDXs act as cytoprotective peroxidases against oxidative stress by reducing peroxides (hydrogen peroxide (H_2_O_2_), alkyl hydroperoxides, and peroxynitrite), as modulators of redox signaling, and as chaperones [1,2,3]. Unlike prosthetic group-dependent enzymes, the PRDX catalytic cycle relies on a conserved Cys, so-called peroxydatic Cys, located in the N-terminal part of the protein. When the thiol (-SH) of this peroxydatic Cys is oxidized by peroxide, this results in a reactive sulfenic acid (-SOH) [4]. Depending on the PRDX subfamily, the sulfenic acid undergoes a condensation reaction with: (i) a thiol on an adjacent subunit to form an inter-subunit disulfide in 2-Cys subfamily (PRDX1–4) [5], (ii) a thiol on the same subunit to form an intra-subunit disulfide in atypical 2-Cys subfamily (PRDX5) [6], or (iii) the π isoform of glutathione-*S*-transferase to form an heterodimeric disulfide in 1-Cys subfamily (PRDX6) [7]. The disulfide is reduced in vivo by a thioredoxin (Trx) for mammalian PRDX1–5 [8], and by glutathione for mammalian PRDX6 [9]. Hyperoxidation can also occur in vivo for typical 2-Cys PRDXs, with the sulfenic acid being oxidized to a sulfinic acid (-SO_2_H) [10], the latter being reduced by the sulfiredoxin system [11], or to an irreversible sulfonic acid (-SO_3_H) [12].

Mammalian PRDXs have also been shown to play key roles extracellularly by acting as damage-associated molecular patterns (DAMPs), triggering the production of cytokines in macrophages [13,14,15,16,17]. In humans, extracellular PRDXs released from necrotic brain cells have notably been shown to worsen brain ischemia-reperfusion injury by inducing expression of proinflammatory cytokines through activation of Toll-like receptor (TLR) 2 (TLR2) and 4 (TLR4) [14]. Among the PRDXs, PRDX5 in particular was shown to be a strong TLR activator [18]. The molecular mechanism underlying this TLR activation remains unclear, although we have recently shown using atomic force microscopy (AFM)-based single-molecule force spectroscopy (SMFS) that recombinant human PRDX5 was able to specifically bind to TLR4 [19]. However, very few studies have focused on the importance of the redox state of catalytic PRDX Cys in their DAMP function [13,14].

Here, we combine an array of interdisciplinary techniques including AFM-based SMFS, size-exclusion chromatography–multiangle light scattering (SEC-MALS), mass spectrometry (MS), enzyme-linked immunosorbent assay (ELISA), flow cytometry (FC), and confocal laser scanning microscopy (CLSM), to investigate the importance of the redox state of human PRDX5 redox-active Cys in the TLR4-dependent pro-inflammatory activity of the protein. By disrupting the C47-C151 disulfide bridge, either through C47 point mutation or dithiothreitol (DTT)/iodoacetamide (IAM) reduction/alkylation treatment, or via hyperoxidation with H_2_O_2_, we demonstrate the role of the C47–C151 disulfide bridge in the PRDX5 interleukin (IL)-8-inducing ability. To determine whether TLR4 is able to discriminate between the PRDX5 redox forms, we used SMFS to evaluate the binding kinetics of reduced, oxidized (disulfide) and hyperoxidized PRDX5 to TLR4. Our results indicate that all redox forms bind TLR4 with a similar kinetic dissociation rate ($k_{off}$), with the exception of hyperoxidized PRDX5 that is not able to establish a specific interaction with the receptor. We further show with a flow cytometry binding assay that the binding propensity of disulfide PRDX5 to the surface of living cells expressing TLR4 is higher than that of the mutant protein. Overall, these results provide the basis for better understanding how the redox state of catalytic redox-sensitive Cys allows PRDXs to promote inflammation in inflammatory-related diseases, which is of crucial importance for the development of new therapeutic strategies related to ischemia-reperfusion injury.

## 2. Materials and Methods

### 2.1. Production and Purification of Recombinant Human PRDX5 and C47S Mutant PRDX5

The expression and purification of recombinant wild-type (dsPRDX5) and C47S mutant (PRDX5^C47S^) *N*-terminal 6xHis-tagged human PRDX5 are described elsewhere [6,19,20]. Briefly, pQE-30 expression vector (Qiagen, Hilden, Germany) coding for wild-type and C47S mutant human PRDX5 were used to transform *Escherichia coli* (*E. coli*) strain M15 (pRep4). *E. coli* were grown at 37 °C in lysogeny broth (LB) medium containing 1 mM isopropyl β-D-1-thiogalactopyranoside (IPTG). Pelleted cells were lysed in 10 mM imidazole, 50 mM phosphate, 300 mM NaCl (pH 8) by sonication and clarified by centrifugation. The supernatant containing 6xHis-tagged PRDX5 was loaded onto a Ni^2+^-nitrilotriacetic acid (NTA) column (Qiagen). The column was washed and the protein was eluted with 50 mM phosphate, 300 mM NaCl, 250 mM imidazole (pH 8). Eluted protein was then dialyzed against phosphate-buffered saline (PBS) (pH 7.2).

### 2.2. Preparation of the PRDX5 Redox Forms

To generate the chemically reduced or hyperoxidized forms, recombinant human PRDX5 (dsPRDX5) was first exposed to 10 mM dithiothreitol (DTT, Sigma-Aldrich, St. Louis, MO, USA) at room temperature (RT) for 1 h to reduce the disulfide bonds. To block the free cysteine thiols in order to maintain the protein in a reduced state, DTT-reduced PRDX5 was subsequently alkylated using 55 mM iodoacetamide [14] (IAM, Sigma-Aldrich) in a dark chamber at RT for 1 h (r/aPRDX5), as described previously. To oxidize cysteine residues to sulfinic or sulfonic acids, DTT-reduced PRDX5 was incubated with 5 mM H_2_O_2_ (Carl Roth, Karlsruhe, Germany) [21] at RT for 1 h (s/sPRDX5).

dsPRDX5, PRDX5^C47S^, r/aPRDX5, and s/sPRDX5 solutions were ultra-filtered using Amicon Ultra-0.5 centrifugal filter devices (Merck, Darmstadt, Germany). First, 500 µL of protein samples were concentrated by spinning the device at 14,000× *g* for 30 min. Then, the filtrate was discarded, and the concentrate was reconstituted to the original sample volume (500 µL) with Dulbecco’s Phosphate Buffered Saline (DPBS, Sigma-Aldrich). This process was repeated six times so that the concentrations of the residual DTT and IAM or H_2_O_2_ were sufficiently reduced (~10^6^ times). To recover the concentrated proteins, the filter device was placed upside down in a clean microcentrifuge tube and spinned for 2 min at 1000× *g* to transfer the sample from the device to the tube. The sample was then reconstituted to the original volume (500 µL).

Protein samples were quantified either using the NanoDrop^TM^ One/One^C^ Microvolume UV-Vis spectrophotometer (ThermoFisher, Waltham, MA, USA) (in MS, SEC-MALS, SMFS and FC experiments) or with the Pierce^TM^ Rapid Gold BCA Protein Assay Kit (Thermo Scientific) (in ELISA and Western blot experiments).

### 2.3. Western Blotting for Detection of Hyperoxidized PRDX5

100 ng of each redox form was loaded on two SDS-polyacrylamide gels (Any kD Mini-PROTEAN TGX Precast Protein gel, Bio-Rad, Hercules, CA, USA) and SDS-PAGE was performed under non-reducing conditions. Proteins were then transferred to nitrocellulose membranes (Amersham Protan 0.45 µm NC, GE Healthcare Life Sciences, Chicago, IL, USA). After transfer, proteins were detected using either an anti-PRDX5 serum (1/200) [22] or an anti-PRDX5-SO_3_ antibody (1/2000) (gift from Prof. Sue Goo Rhee, Yonsei University College of Medicine, Seoul, South Korea) [21]. Polyclonal goat anti-rabbit IgG/HRP (1/2000) (Agilent, Santa Clara, CA, USA) was used as the secondary antibody, and BM Chemiluminescence Blotting Substrate (Roche, Basel, Switzerland) was used to visualize the protein bands on an Amersham Imager 600 (GE Healthcare Life Sciences).

### 2.4. Mass Spectrometric Characterization of the PRDX5 Cysteine Redox State

#### 2.4.1. Blocking of Cysteine Redox States

Prior to MS analysis, buffer exchange was accomplished using Amicon Ultra-0.5 centrifugal filter devices (Merck, Darmstadt, Germany) by concentrating the protein samples, discarding the filtrates, and reconstituting the concentrate protein samples to the original volume with 100 mM triethylammonium bicarbonate (pH 8.5) buffer (TEAB, Sigma-Aldrich). Potent cysteine sulfenic acids were trapped with 5 mM 5,5-dimethyl-1,3-cyclo-hexanedione (dimedone, Sigma-Aldrich) for 30 min at RT [23,24], and thiols remaining after redox treatments were subsequently blocked with 55 mM 2-chloroacetamide (CAM) for 30 min at RT in the dark. To be able to determine thioethers formed after alkylation with IAM from those formed after alkylation with CAM, MS analysis of r/aPRDX5 was performed with and without CAM treatment.

#### 2.4.2. Sample Preparation

1 µg of samples was acidified with 1% (*v*/*v*) trifluoroacetic acid (TFA), desalted and concentrated with C4 ZipTip (Millipore, Burlington, MA, USA) according to the manufacturer’s protocol. Samples were then eluted with 10 µL of 50% (*v*/*v*) acetonitrile, 0.1% formic acid.

#### 2.4.3. ESI-Q-TOF Analysis of Intact Proteins by Direct Infusion

Intact protein analysis was done on a SYNAPT G2-Si high definition mass spectrometer (Waters) equipped with a NanoLockSpray dual electrospray ion source (Waters, Milford, MA, USA). Coated fused silica PicoTipTM Econo12 Emitters for nanoelectrospray, outer diameters: 1 µm Tip (New Objective, Littleton, MA, USA) were filled with 5 µL of samples and placed on the Universal NanoFlow™ Sprayer (Waters). The eluent was sprayed at a spray voltage of 1.2 kV. The source temperature was set to 120 °C. The cone gas flow was 20 liters·h^−1^ with a nanoflow gas pressure of 0.3 bar. MS spectra were acquired and processed with MassLynx software (Waters).

#### 2.4.4. Maximum Entropy Deconvolution

MaxEnt1 deconvolution algorithm was used to extract the exact masses with the following parameters: ranges from 17,000 Da to 39,000 Da, resolution of 0.5 Da/channel, Uniform Gaussian peak width at half height of 0.4 Da. A centering process was performed using the centroid method to obtain accurate average molecular masses. The minimum peak width at half height was set to 2, and the fraction of the resolved portion of the peak used to calculate the centroid was set to 80%. The height of a bar represents the sum of intensities of the points across the peak in the continuum trace.

### 2.5. SEC-MALS Analysis of the Oligomeric State

SEC-MALS experiments were performed using an LC-20 Prominence BioInert HPLC system (Shimadzu, Kyoto, Japan), coupled to a miniDAWN TREOS-II MALS instrument (Wyatt Technology, Santa Barbara, CA, USA), a RID-20A refractive index detector (Shimadzu) and an SPD-20A UV-VIS detector (Shimadzu). For chromatographic separation, 50 µL of sample were loaded alternately onto a Superdex 200 Increase 10/300 GL (Cytiva, Marlborough, MA, USA) and a Superdex 75 Increase 10/300 GL (Cytiva) (data not shown) equilibrated in the PBS running buffer (10 mM Na_2_HPO_4_, 1.8 mM KH_2_PO_4_, NaCl 137 mM, KCl 2.7 mM pH 7.4). Data were analyzed with ASTRA 7.3 software (Wyatt Technology) and molecular mass was calculated using a Zimm fit model.

Relative quantification of the absolute molar masses of PRDX5 redox form species was performed by taking the ratio of the calculated mass of the selected peak over the total calculated mass of all peaks of interest, expressed as a percentage.

### 2.6. Generation and Culture of HEK293 Cell Line Expressing TLR4

HEK293 cells expressing human MD-2 and CD14 (HEK293 MD2-CD14) (InvivoGen, San Diego, CA, USA) were plated at a density of ~2 × 10^6^ cells in 100 mm culture dishes and stably transfected with 10 µg pcDNA3-TLR4-YFP (Addgene, Watertown, MA, USA, plasmid #13018) using 25 µL Lipofectamine LTX (ThermoFisher) and 10 µL PLUS Reagent (ThermoFisher) in 1 mL of Opti-MEM (ThermoFisher), when cells reached ~70% confluency. Stable transfectants (HEK293 MD2-CD14 TLR4) were first selected using the antibiotic Geniticin (G418) (InvivoGen). To control TLR4 expression at the cell surface, cells were further sorted based on the fluorescence of an APC-conjugated anti-TLR4 antibody (ThermoFisher) using fluorescence-activated cell sorting (FACS). Single cells were then isolated to produce stable cell lines.

Cells were cultured in Dulbecco’s modified Eagle’s medium (DMEM) GlutaMAX (ThermoFisher) supplemented with 10% (*v*/*v*) heat-inactivated fetal bovine serum (FBS) (Lonza), 100 units·mL^−1^ streptomycin and 100 units·mL^−1^ penicillin (Sigma-Aldrich), 100 mg·mL^−1^ of Normocin (InvivoGen) and 0.5 mg·mL^−1^ G418 (InvivoGen). Cells were grown at 37 °C in a humidified atmosphere supplemented with 5% CO_2_.

### 2.7. Quantification of Human IL-8 Secretion

Prior to performing ELISAs, endotoxins were removed from the PRDX5 productions using the Pierce High Capacity Endotoxin Removal Spin Columns (ThermoFisher), following the manufacturer’s instructions. For each experiment (ELISA), a new redox treatment for r/aPBS, r/aPRDX5, s/sPBS, and s/sPRDX5 was performed, and two independent PRDX5^C47S^ production batches were used.

HEK293 MD2-CD14 TLR4 cells were counted and cultured in 96-well plates with complete growth medium for 48 h before being exposed to 2 µM of the different PRDX5 redox forms, 2 µM HMGB1, and 100 ng·mL^−1^ LPS, or to increased concentrations (1, 2, 5, 10, 15, and 20 µM) of the different PRDX5 redox forms, in DMEM supplemented with 100 units·mL^−1^ streptomycin and 100 units·mL^−1^ penicillin without FBS. Culture medium was harvested 16 h after cell exposure and IL-8 was measured using ELISA kit (BD Biosciences, Franklin Lakes, NJ, USA), following manufacturer’s instructions.

### 2.8. Functionalization of AFM Tips

PRDX5 molecules were grafted onto the AFM tips (MST probes, Bruker, Billerica, MA, USA) using (3-aminopropyl) triethoxysilane (APTES) coating and a heterobifunctional polyethylene glycol (PEG_24_) linker composed of an aldehyde molecule at one end and an *N*-hydroxy succinimide (NHS) molecule at the other end. The protocol was adapted from Professor H. J. Gruber (Johannes Kepler University, Institute of Biophysics, Linz, Austria).

#### 2.8.1. Amino-Functionalization

Probes were first washed three times in chloroform, dried with a gentle stream of nitrogen and cleaned for 15 min in an ultraviolet radiation and ozone (UV-O_3_) cleaner (JeLight, Irvine, CA, USA). Probes were placed for 2 h in a hermetic desiccator filled with argon, in which were placed a tray containing 30 µL of APTES and another tray containing 10 µL of triethylamine. Finally, the trays were removed and probes were kept for 2 more days in the desiccator filled with argon.

#### 2.8.2. Coupling of Aldehyde-PEG_24_-NHS Linker

The amino-functionalized probes were first immersed for 2 h in a solution containing 1 mg of aldehyde-PEG_24_-NHS linker dissolved in 0.5 mL of chloroform, and 30 µL of triethylamine. Probes were then washed three times in chloroform and dried with a gentle stream of nitrogen. The probes were immersed in 100 µL of 0.2 mg·mL^−1^ PRDX5 solution and 2 µL of a freshly prepared solution of 1 M NaCNBH_3_ (with 20 mM NaOH) for 1 h. The reaction was quenched by adding 5 µL of 1 M ethanolamine solution (pH 8). After 10 min, the tips were rinsed three times with phosphate buffered saline (PBS) and stored in PBS. Experiments were conducted within 48 h following protein coupling.

### 2.9. Preparation of TLR4-Coated Surfaces

Recombinant human TLR4 (R&D Systems, Minneapolis, MN, USA) was covalently attached to a mixed self-assembled monolayer (SAM) of carboxyl (COOH)-and hydroxyl (OH)-terminated alkanethiols immobilized onto model surfaces using NHS/*N*-(3-dimethylaminopropyl)-*N*′-ethylcarbodiimide hydrochloride (EDC) chemistry.

#### 2.9.1. Alkanethiol SAM Formation

Gold-coated silicon surfaces of ~1 × 1 cm were first washed with ethanol, dried with a gentle flow of nitrogen, and cleaned for 15 min by UV-O_3_ treatment. Surfaces were then immersed overnight in an ethanol solution containing a 1 mM mixture (1:99 mol/mol) of 16-mercaptohexadodecanoic acid (HS(CH_2_)_16_COOH) (Sigma-Aldrich) and 11-mercapto-1-undecanol (HS(CH_2_)_11_OH; Sigma-Aldrich).

#### 2.9.2. NHS/EDC Chemistry

After sonicating briefly to remove alkanethiol aggregates that may be adsorbed, surfaces were rinsed with ethanol and dried with nitrogen. Next, surfaces were immersed for 30 min in a solution of 175 mM NHS (Sigma-Aldrich) and 260 mM EDC (Sigma-Aldrich) in ultrapure water and rinsed 10 times with PBS. The activated surfaces were finally incubated with 0.1 mg·mL^−1^ TLR4 in PBS for 2 h and rinsed three times in PBS. Experiments were conducted within 48 h following surface functionalization.

### 2.10. FD-Curve Based AFM on TLR4-Coated Surfaces

SMFS on model surfaces was operated with a NanoScope 8 Multimode AFM (Bruker) and with the NanoScope 9.2 software (Bruker), using PRDX5-functionalized MSCT tips (C triangular cantilevers, Bruker) and TLR4-functionalized gold surfaces in PBS at ~25 °C. Using Force Volume in contact mode, FD curves were recorded on 5 × 5 µm^2^ arrays of 32 × 32 curves. Constant approach speed was set at 1 µm·s^−1^ and retraction speeds were varied (100 nm·s^−1^, 200 nm·s^−1^, 500 nm·s^−1^, 1 µm·s^−1^, 2 µm·s^−1^, 10 µm·s^−1^). A ramp size of 200 nm and a force threshold of 300 pN were used. Cantilever spring constants were calibrated on mica surfaces with the thermal noise method [25].

Force curves were analyzed using the Nanoscope Analysis 1.8 software (Bruker). FD curves were first extracted from individual adhesion maps and sorted to keep the curves displaying specific adhesion event(s). For each tip retraction speed, the binding frequency (BF) was calculated by dividing the number of specific adhesive curves by the total number of FD curves recorded for each map. For the DFS analysis, the baseline of the retraction curves was first corrected using a linear fit to remove tilt and offset. Rupture forces (expressed in pN) were extracted by measuring the height of the adhesive peak(s), and LRs (expressed in pN·s^−1^) were extracted by fitting the adhesive peak on the force-time curve just before rupture with a linear model. Determination of the kinetic parameters of the interactions was performed on OriginPro 9.8.0.200 software (OriginLab Corporation, Northampton, MA, USA). The spectra of LRs were divided into five or six discrete ranges (LR#1: 315–1000 pN·s^−1^; LR#2: 1000–3150 pN·s^−1^; LR#3: 3150–10,000 pN·s^−1^; LR#4: 10,000–31,500 pN·s^−1^; LR#5: 31,500–100,000 pN·s^−1^; LR#6: 100,000–315,000 pN·s^−1^), depending on the redox form. For each LR range, a Kernel distribution was used to represent the probability density function (pdf) of the rupture forces, based on the following function:$$ksdensity\;\left(x,Vx,w\right)=\frac{1}{n}\overset{n}{\underset{i=1}{\sum }}\frac{1}{\sqrt{2\pi }w}e^{\frac{-{\left(x-vX_{i}\right)}^{2}}{2w^{2}}}$$

This function returns the kernel density at x for a given vector Vx of size n with a bandwidth w, calculated as follows:$$w=0.9min\left(\hat{\sigma },\frac{IQR}{1.349}\right)n^{-1/5}$$
where $\hat{\sigma }$ and IQR are the standard deviation and interquartile range of Vx, respectively. The second derivative of the Kernel Smooth function was used to search for hidden positive peaks, by choosing the lower peak filtering threshold allowing the fit to converge. Peaks were then fitted with Gaussian functions using the Levenberg-Marquardt algorithm. The peak centers determined the most probable rupture forces for single and multiple interactions. The Bell-Evans model [26] was fitted to most probable rupture forces corresponding to single interactions to determine the dissociation rate constant $k_{off}$ and the distance to the transition state $x_{\beta }$:$$F=\frac{k_{B}T}{x_{\beta }}\left(ln\left(LR\right)+ln\left(\frac{x_{\beta }}{k_{B}Tk_{off}}\right)\right)$$
where F is the rupture force, $k_{B}T$ is the thermal energy, and LR is the loading rate. The predicted rupture forces for two (n = 2) and three (n = 3) simultaneous uncorrelated interactions were determined using Williams-Evans prediction [27]:$$LR=k_{off}\frac{k_{B}T}{x_{\beta }}{\left(\overset{N}{\underset{n=1}{\sum }}\frac{1}{n^{2}}exp\left(\frac{-Fx_{\beta }}{nk_{B}T}\right)\right)}^{-1}$$

### 2.11. Confocal Microscopy Analysis of PRDX5 Binding to the Cell Surface

Prior to imaging, HEK293 MD2-CD14 TLR4 cells were seeded onto polyethylenimine (PEI)-coated glass bottom μ-dishes and left overnight to grow to confluence. 100 µM NTA-FITC (Santa Cruz Biotechnology, Dallas, TX, USA) were incubated with 500 µM NiCl_2_ for 1 h at RT, and 10 µM dsPRDX5 was then added for 1 h. Unbound probe was then removed using a PD SpinTrap G-25 column (Cytiva), by following manufacturer’s instructions. Cells were imaged using a Zeiss 980 confocal laser scanning microscope (Zeiss GmbH, Oberkochen, Germany). This microscope is equipped with a 405 diode and Helium-Neon lasers, GaAsP detectors, and additional modifications to allow for long term live cell imaging. These modifications include a chamber that maintains cells at 37 °C, with 5% CO_2_ in air, and a relative humidity of >95%. Excitation and emission settings were determined using YFP positive cells, prior to the addition of PRDX5-NTA-FITC, to reduce crosstalk between the channels. Excitation for FITC and YFP were at 488 and 514 nm respectively, with the key collection bands being between 495–522 nm for FITC and 565–648 nm, with the channels being collected sequentially.

### 2.12. Flow Cytometry Binding Assay

Cells were first harvested with Accutase solution (Sigma-Aldrich) prior to performing cell count and viability analysis. Cells were resuspended in ice-cold medium and ~5 × 10^5^ cells·mL^−1^ were exposed to 1 µM of the PRDX5 redox forms or to increase concentration of dsPRDX5 in DMEM for 40 min on ice to prevent endocytosis. As negative control, cells were incubated with medium only. Cells were centrifuged at 500× *g* for 5 min at 4 °C and then incubated with PBS containing 0.2% BSA for 15 min on ice. Cells were resuspended in anti-PRDX5 serum (1/200) for 40 min on ice, before being washed three times with PBS/0.2% BSA. Cells were finally labeled for 40 min on ice with goat anti-rabbit IgG secondary antibody conjugated to Alexa Fluor 647 Plus (1/250) (ThermoFisher), washed three times, and resuspended in PBS. Samples were acquired on the Guava easyCyte System (Merck) using GuavaSoft software. Gates were set to select living Alexa Fluor 647 Plus-positive cells. Median fluorescence intensity (MFI) of control cells was subtracted from MFI of PRDX5-exposed cells.

### 2.13. Statistical Analysis

JMP Pro 15.2.0 software (SAS Institute, Cary, NC, USA) was used to perform the statistical tests. For ELISAs, FC experiments and SMFS BF measurements, a mixed model was used with the conditions (either exposition agents in ELISAs, or PRDX5 redox forms in FC and SMFS) as fixed independent variables and replicates as a random variable defining a random intercept. When the condition was significant (*p*-value < 5%), pairwise comparisons were computed with a Tukey HSD test. To fulfill the assumptions of the model, a logarithmic transformation (log_10_) of the response (IL-8 concentration) was performed in ELISAs.

Bell-Evans fitting results were compared with OriginPro 9.8.0.200 software. F-test was used to determine whether pairwise datasets were significantly different from each other. A *p*-value was then calculated based on the F value.

## 3. Results and Discussion

### 3.1. Production and Characterization of PRDX5 Redox Forms

To evaluate the influence of the redox state of human PRDX5 catalytic cysteine residues (Cys) on its DAMP function, we used four different redox forms (Figure 1). Purified recombinant human 6xHis-tagged PRDX5 produced in *Escherichia* coli (*E. coli*) has a disulfide bridge between the C47 and the C151 [8,28] (dsPRDX5) (Figure 1A,B). A mutated form of the protein was obtained by replacing the C47 by a serine (S), which maintains the protein in a reduced state by preventing the formation of the C47-C151 disulfide bridge (PRDX5^C47S^) (Figure 1A,B). Previous studies showed that the 6xHis tag does not alter the enzymatic activity of the protein, which remains active [29,30,31,32,33,34,35]. To alter PRDX5 redox state without modifying the structure, purified PRDX5 proteins were also chemically treated with dithiothreitol (DTT) to reduce disulfide bonds. Pre-reduced proteins were then exposed either to the alkylating agent iodoacetamide (IAM) to prevent their re-oxidation (r/aPRDX5, Figure 1B), or to a high concentration of H_2_O_2_ to form Cys sulfinic and/or sulfonic acids (s/sPRDX5) (Figure 1B). It is important to note that PRDX5 is known to be highly resistant to hyperoxidation compared to typical 2-Cys PRDXs, and that, to our knowledge, hyperoxidized PRDX5 has never been detected in vivo [21]. Therefore, the implication of PRDX5 hyperoxidized forms in physiological or pathophysiological mechanisms is very unlikely, however s/sPRDX5 serves here as an important comparison. We pre-treated PRDX5 with DTT before hyperoxidation, as per Yang et al. [10], as H_2_O_2_ alone is not sufficient for disulfide disruption.

Next, we characterized the PRDX5 redox forms using various approaches (Figure 2). Western blot analysis (Figure 2A) first revealed that dsPRDX5 showed similar bands before and after ultrafiltration, indicating that the ultrafiltration process used to remove the residual agents after the chemical treatments does not alter the proteins. We then confirmed that DTT/H_2_O_2_ treatment led to at least partial hyperoxidation of PRDX5 Cys by comparing similar blots probed either with an antiserum raised against recombinant human PRDX5 or an antibody highly specific for the sulfonated PRDX5 form [21].

We further characterized the redox state of the PRDX5 redox forms using direct infusion electrospray ionization quadrupole time-of-flight mass spectrometry (ESI-Q-TOF-MS) on intact proteins (Figure 2B). After redox treatments and prior to MS analysis, potent Cys sulfenic acids were trapped with dimedone, and thiols were subsequently blocked by alkylation with 2-chloroacetamide (CAM). Raw multiply-charged electrospray spectra were processed with the maximum entropy method [36,37,38] to produce true molecular mass spectra (Appendix A).

Based on the amino acid sequence of dsPRDX5 and PRDX5^C47S^, the calculated mass of the proteins with reduced Cys thiols are 18,297.81 and 18,281.75 Da, respectively. The deconvoluted mass spectra revealed major monomeric and minor dimeric species for each redox form (Appendix A). The centroid spectrum of dsPRDX5 shows four major peaks related to the monomeric species (Figure 2B, Appendix A), of which mass and identification are presented in Appendix A. Peaks #1 and #3 correspond to the protein with a disulfide bond (mass shift of -2 Da), without or with an additional carbamidomethyl group (mass shift of +57 Da), respectively. The two secondary peaks (#2 and #4) indicate a mass increase of 14 Da compared to the parent peaks, and are most likely due to a *N*-terminal methylation [39]. This phenomenon has been observed by other researchers who performed similar intact-protein ESI-MS analysis on the recombinant *N*-terminal 6xHis-tagged Z domain of protein A [40]. The deconvoluted mass spectrum of PRDX5^C47S^ reveals two major peaks for the monomeric species (Figure 2B, Appendix A). The parent peak (#1) corresponds to the mutant protein with a serine residue (Ser) replacing the Cys and a carbamidomethyl group, and the secondary peak (#2) indicates an additional mass of 14 Da, as for dsPRDX5. In the case of r/aPRDX5, the two major peaks detected correspond to the protein with three carbamidomethyl groups, either with or without a 14-Da increase in mass (Figure 2B, Appendix A). As both IAM and CAM modifications result in the covalent addition of a carbamidomethyl group (mass shift of +57 Da), we performed the analysis without CAM alkylation in order to determine which adducts were formed after IAM redox treatment (insets in Appendix A). The analysis performed without CAM reveals identical peaks with a mass shift of −57 Da each, indicating that alkylating the protein with IAM at pH 7.2 more likely forms thioethers on the two more reactive Cys, i.e., the ones involved in the disulfide bond. Lastly, the major peaks resolved on the deconvoluted mass spectrum of s/sPRDX5 (peaks #1, #2, #5, and #7) are identical to those of dsPRDX5 (Figure 2B, Appendix A). However, several other peaks (#3, #4, #6, #8, #9, #10, and #11) were detected and indicate that hyperoxidizing the DTT-reduced protein with H_2_O_2_ results in a mix of species composed of Cys sulfinic acid (-SO_2_H), sulfonic acids (-SO_3_H), and carbamidomethyl groups. Summing the peak intensities corresponding to these species gives a proportion of ~42%, but the disulfide bond remains in many of these species, which is in agreement with the fact that PRDX5 is resistant to hyperoxidation by H_2_O_2_ [21]. The identification of these peaks is presented in Appendix A. No sulfenic intermediate was identified (mass shift of +138 Da). Moreover, analysis of the fragments corresponding to peptides containing C47 using LC-Q-TOF-MS/MS revealed the presence of the redox modifications (disulfide bridge, carbamidomethyl group, sulfinic acid, sulfonic acid) on the Cys (Appendix A). Overall, these results demonstrate that dsPRDX5, PRDX5^C47S^, and r/aPRDX5 are largely homogenous redox forms and confirm the efficiency of the reducing/alkylating treatment. However, s/sPRDX5 is only partially hyperoxidized, with a proportion of species having a disulfide bridge.

While the oligomeric conformation of 2-Cys PRDX subfamily directly links redox state to function [41,42], atypical 2-Cys PRDX5 is thought to exist as a dimer irrespective of the redox state of the cysteine residues [8,43]. Western blot performed under non-reducing conditions revealed the presence of monomers, but also of dimers (Figure 2A), and to a lesser extent, of oligomers (Appendix A). Although these results are consistent with previous studies [35,44], whether covalent dimers (and oligomers) linked by intramolecular disulfide bridges are physiologically relevant remains to be determined [43]. We further refined the oligomeric state picture of PRDX5 for all redox states in native conditions using size-exclusion chromatography–multiangle light scattering (SEC-MALS) (Figure 2C, see spectra in Appendix A). The main quaternary structure detected for all conditions was the dimeric form. In dsPRDX5, however, a relatively high fraction (~30%) of monomers could also be identified, and r/aPRDX5 shows ~15% of proteins in a monomer-dimer equilibrium. This is consistent with previous results showing that there is a rapid equilibrium between a major dimer and a minor monomer species [44]. A very small fraction of tetramers was observed for PRDX5^C47S^ and s/sPRDX5. Although the covalent or non-covalent nature of such dimers and oligomers cannot be determined in SEC-MALS, MS analysis of intact proteins revealed the presence of non-covalent dimers for PRDX5^C47S^ and r/aPRDX5 (Appendix A). This suggests that such non-covalent dimers are also present in dsPRDX5 and s/sPRDX5. Overall, these data indicate that the redox state of PRDX5 does not alter its oligomeric conformation, facilitating further analysis and interpretation.

### 3.2. The PRDX5 Disulfide Bond Allows a Stronger Cytokine-Stimulating Activity

Our previous study showed that dsPRDX5 was able to stimulate IL-1β secretion from PMA-differentiated THP-1 cells [19]. To demonstrate the direct role of TLR4 in PRDX5-triggered cytokine secretion, we examined the IL-8 production from HEK293 MD2-CD14 transfected with TLR4 cells using ELISA (Figure 3). We first used FC to validate the stable transfection and to confirm that the receptor was expressed at the cell surface (Figure 3A). We also established that the receptor was functional by exposing the cells to LPS, which requires both MD-2 and CD14 to trigger cytokine secretion [45,46,47] (Figure 3C). LPS exposure resulted in a secretion of 14,085 ± 2401 pg·mL^−1^ (mean ± SD_pooled_, *N* = 3).

We then verified that the residual DTT, IAM and H_2_O_2_ remaining after protein redox treatments had been efficiently removed by ultrafiltration and thus would not impact cell viability or interfere with IL-8 secretion. To do so, we treated PBS with the same reducing and hyperoxidizing agents that we used for dsPRDX5, and we ultra-filtrated the samples. As shown in Figure 3B, exposing cells to r/aPBS or s/sPBS resulted in similar IL-8 secretion levels (175 ± 17 and 174 ± 17 pg·mL^−1^, respectively, mean ± SD_pooled_, *N* = 3) as with PBS (187 ± 61 pg·mL^−1^, mean ± SD_pooled_, *N* = 3), confirming that the further results would not be on account of residual treatment agents.

We next assessed the IL-8 production from HEK293 MD2-CD14 TLR4 cells mediated by exposure to the PRDX5 redox forms (Figure 3C). We observed that dsPRDX5 yielded higher IL-8 stimulating activity with 4608 ± 288 pg·mL^−1^ (mean ± SD_pooled_, *N* = 3) compared to the other redox forms. The C47S point mutation led to significantly reduced IL-8 secretion level (893 ± 142 pg·mL^−1^, mean ± SD_pooled_, *N* = 3). The same tendency could be observed with the chemically reduced/alkylated protein, albeit less prominent (2437 ± 343 pg·mL^−1^, mean ± SD_pooled_, *N* = 3). Similarly, hyperoxidizing the protein in vitro with H_2_O_2_ resulted in a significant decrease in IL-8 secretion (1447 ± 136 pg·mL^−1^, mean ± SD_pooled_, *N* = 3) compared to dsPRDX5. As control, we used another DAMP protein that can signal through TLR4, namely high-mobility group box 1 (HMGB1) [48], which led to an IL-8 concentration of 19,227 ± 2695 pg·mL^−1^ (mean ± SD_pooled_, *N* = 3). We also demonstrated that the magnitude of IL-8 secretion was dependent on the concentration of the redox forms (Figure 3D). Our results highlight the importance of the C47-C151 disulfide bond in PRDX5-mediated cytokine production through TLR4, and is in line with other studies showing that the activity of HMGB1 strongly depends on the redox state of its three Cys [49,50]. Nevertheless, our results are in contradiction with a previous study showing that the disruption of PRDX5 Cys by point mutations or IAM treatment did not affect the IL-23p19 mRNA expression level in bone marrow-derived dendritic (BMDC) cells [14]. However, their conditions differ from ours: (i) PRDX5 was reduced with only 1 mM DTT, while several studies use 10 mM [21,29]; (ii) they investigated the mRNA level, although they showed for PRDX1 and PRDX2 that IL-23p19 mRNA expression level does not correlate with IL-23 expression; (iii) BMDC cells were exposed for 1 h, while we treated our cells for 16 h, (iv) BMDC cells have a wider TLR expression profile [51], making it difficult to interpret the contribution of TLR4. The effect of the Cys redox state has also been studied for PRDXs from the typical 2-Cys subfamily, with conflicting results. While Riddell et al. showed that human PRDX1 interaction with TLR4 was independent of its peroxidase activity [13], López et al. recently demonstrated that pro-inflammatory properties of the cytosolic tryparedoxin peroxidase from *Trypanosoma cruzi* depend on the presence of peroxidatic cysteine [52].

### 3.3. Disulfide, Mutant and Reduced PRDX5 Bind to TLR4 with Similar Kinetic Off-Rates

We previously demonstrated with AFM-based SMFS that dsPRDX5 was able to specifically interact with TLR4 [19]. To determine whether the difference in the cytokine-inducing capability of the PRDX5 redox forms could originate from differential binding properties to TLR4, we used AFM-based SMFS to probe the interaction between the PRDX5 redox forms and TLR4 (Figure 4A). To this end, we covalently grafted the PRDX5 redox forms to AFM tips by means of a flexible polyethylene glycol (PEG_24_) linker, and we covalently attached TLR4 molecules to a mixed self-assembled monolayer of carboxyl- and hydroxyl-terminated alkanethiols immobilized onto an inert gold surface using *N*-hydroxysuccinimide (NHS)/*N*-(3-dimethylaminopropyl)-*N*′-ethylcarbodiimide hydrochloride (EDC) chemistry. We then used force-distance (FD) curve-based AFM to investigate the binding kinetics of the PRDX5 redox forms (Figure 4B, Appendix A). We recorded multiple maps (30 for dsPRDX5, PRDX5^C47S^ and r/aPRDX5; 35 for s/sPRDX5) at different tip retraction speeds (100 nm·s^−1^, 200 nm·s^−1^, 500 nm·s^−1^, 1 µm·s^−1^, 2 µm·s^−1^, and 10 µm·s^−1^).

We first assessed the binding frequency (BF) by examining the FD curves showing specific adhesive events between the tip-bound PRDX5 and surface-immobilized TLR4 (i.e., whose extension patterns could be fitted with the worm-like chain model) (Figure 4C, Appendix A). The BF is given by the ratio between the number of specific events and that of total recorded events [53]. There was no significant difference between the different redox forms, except for the r/aPRDX5 that showed a higher BF.

To investigate the binding kinetics of PRDX5-TLR4 interactions for the different redox forms, we performed a dynamic force spectroscopy (DFS) analysis. For each specific adhesive curve, we extracted the rupture force (F) and the loading rate (LR), which corresponds to the force applied over time (Figure 4B). We divided the spectrum of LRs probed into discrete ranges (LR#1–6) as performed in previous studies [19,54,55], and for each of them, we used a Kernel distribution of the rupture forces to define the most probable forces for single and multiples interactions (Appendix A). The second derivative of the density function allowed finding hidden peaks, which were fitted with Gaussian functions (Figure 4D, Appendix A). The resulting Gaussian function overlapped very well the Kernel distribution.

With the exception of s/sPRDX5, all the DFS plots showed a linear dependency of the most probable rupture forces with the LR for single interactions, which is frequently observed for ligand-receptor interactions and is explained by a single free-energy barrier being crossed during the mechanical pulling [56]. The Bell-Evans model [26] was used to fit to the most probable forces measured for the single interaction at the different LR ranges. From the fit (Figure 4D, dark line), we extracted the dissociation rate constant ($k_{off}$), which corresponds to the inverse of the interaction lifetime, as well as the distance to the transition state ($x_{\beta }$), which provides useful information related to the chemistry of interaction. The $k_{off}$ values obtained were 1.02 ± 1.57 s^−1^, 1.97 ± 0.33 s^−1^, and 2.47 ± 2.29 s^−1^ for dsPRDX5, PRDX5^C47S^, and r/aPRDX5, respectively. Regarding s/sPRDX5, the DFS plot reveals no increase in the binding force with the LR, suggesting an unspecific interaction. A similar $x_{\beta }$ was observed for dsPRDX5 (0.57 ± 0.18 nm), PRDX5^C47S^ (0.44 ± 0.02 nm) et r/aPRDX5 (0.46 ± 0.10 nm), indicating a similar binding conformation. Using these parameters, we applied the Williams-Evans prediction [57] to infer the interaction forces of multiple uncorrelated interactions breaking simultaneously (Figure 4D, light lines). These predictions are in good agreement with the mean rupture forces observed for double and triple interactions. By performing a F-test on the Bell-Evans fittings, we could determine that dsPRDX5 and PRDX5^C47S^, as well as dsPRDX5 and r/aPRDX5, were not statistically distinct, with 0.05 significance level (*p*-value of 0.50492 and 0.63374, respectively). This can be visualized by superimposing the DFS plots of the different redox forms, as shown in Appendix A. The residual dsPRDX5 present in the hyperoxidized sample can explain the higher rupture forces observed, which probably result from the interaction between dsPRDX5 and TLR4. Overall, these results indicate that the differences observed in the IL-8-stimulating activity of the different redox forms cannot be attributable to differential interaction lifetime with TLR4. Further work is needed to investigate the associate rate constant ($k_{on}$) in order to determine whether TLR4 alone is sufficient to discriminate between the redox forms.

### 3.4. PRDX5^C47S^ Shows a Lower Propensity to Bind to the Cell Surface

We consequently turned to living HEK293 cells expressing TLR4 to investigate the binding of PRDX5 molecules in a cellular context. We first performed a colocalization experiment to visually confirm that dsPRDX5 binding to TLR4 is responsible for the inflammatory response (Figure 5A). To this end, we incubated HEK293 MD2-CD14 TLR4 cells with PRDX5 labeled with NTA-FITC using Ni^2+^ (PRDX5-NTA-FITC). Shortly after the addition of PRDX-NTA-FITC, a membrane staining motif began to emerge, with discrete pockets of FITC fluorescence present at the cell membrane. Of particular note, fluorescence signals present in filopodia and focal adhesions were commonly co-localized. This observation was subsequently followed by FITC positive vesicles gathering at the cell periphery and trafficking within the cell interior. Together these staining behaviors indicate binding and internalization. To ensure that the fluorescence observed was not merely crosstalk, λ-scans were performed to assess whether the fluorescent species present included the earlier emission peak from the FITC dye and fluorescence intensities across the membrane examined using a simple Line of Response (Appendix A).

We further investigated the binding level of redox PRDX5 isoforms to the surface of HEK293 MD2-CD14 TLR4 cells using a FC binding assay. To prevent endocytosis, cells were incubated with PRDX5 on ice, and labelled with an antiserum against PRDX5. As a control, we also labelled untreated cells. We first showed that dsPRDX5 binding to the surface of cells expressing TLR4 was concentration-dependent (Figure 5B). These results confirm the role of TLR4 in the proinflammatory activity of PRDX5 and demonstrate that LPS is not responsible for the concentration-dependent increase in IL-8 secretion following exposure to PRDX5 (Figure 3D). At identical concentration, PRDX5^C47S^ shows a ~3-fold decrease in binding propensity compared to dsPRDX5 (Figure 5B). However, this result was not observed for r/aPRDX5 and s/sPRDX5. As the difference observed between dsPRDX5 and the mutant protein was not observed with SMFS, this result could indicate either that TLR4 conformation (i.e., dimer) and/or stabilization in the plasma membrane is important for the binding, or that the TLR4 accessory molecules MD-2 and/or CD14 might allow discriminating between redox forms. Further investigation is needed to understand the role of TLR4 accessory molecules in this context.

## 4. Conclusions

Human PRDX5 has been shown to enhance brain ischemia-reperfusion injury through activation of TLR2- and TLR4-mediated inflammation. Although our previous findings reported that PRDX5 was able to directly interact with TLR4, the role of the redox state of PRDX5 cysteine residues remained to be investigated. Indeed, PRDX5 is a redox-sensitive protein that act as a peroxidase inside the cell. Here, we investigated the impact of the PRDX5 redox state on its TLR4-dependent proinflammatory activity by modifying the redox state of PRDX5 cysteine residues, either by point-mutation or by chemical treatments. Using molecular biology and biophysics approaches, we provide evidence that the disulfide bridge between PRDX5 enzymatic cysteine is required to allow a strong TLR4-dependent IL-8 secretion. Although the interaction lifetime between TLR4 and oxidized or reduced PRDX5 is not different, binding assays performed on living cells expressing TLR4 demonstrated that disulfide PRDX5 has a higher propensity to bind the cell surface than the mutant protein lacking the peroxidatic cysteine. Taken together, these results shed new light on the importance of the protein redox biology in inflammation, which is of crucial importance for the development of therapeutic treatments to administer after a stroke.

## Figures and Tables

**Figure 1 antioxidants-10-01902-f001:**
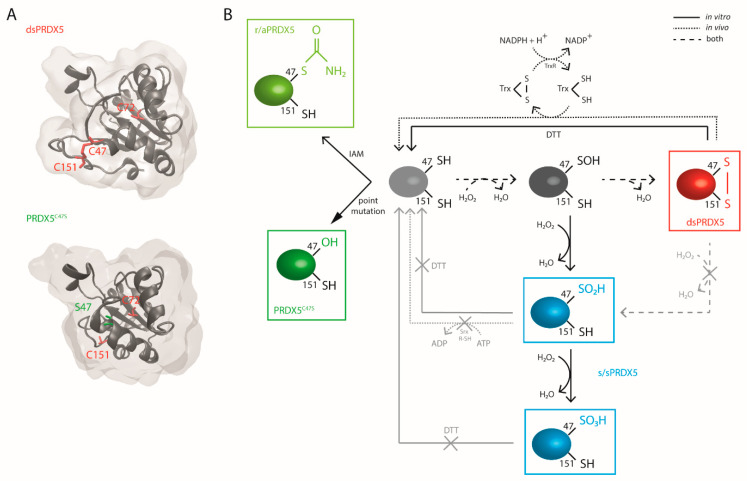
Structural and redox state insights of PRDX5. (**A**) Crystal structures of recombinant human PRDX5 (upper, PDB ID: 2vl3) and mutated C47S PRDX5 (lower, PDB ID: 1urm), showing that changes in Cys redox state is accompanied by conformational changes. dsPRDX5 contains three Cys (C47, C72, C151), with two of them being catalytically active (C45 and C151). (**B**) Schematic of the PRDX5 C47 redox modifications. The PRDX5 C47 thiol (-SH) can be oxidized in vitro or in vivo to an intermediate sulfenic acid (-SOH) by H_2_O_2_ or other peroxides. The reactive -SOH either engages into an intramolecular disulfide bond with C151 (dsPRDX5, red) or is hyperoxidized in vitro to sulfinic acid (-SO_2_H) or further to sulfonic acid (-SO_3_H) (s/sPRDX5, blue). In vivo, the disulfide bridge is reduced by thioredoxin (Trx). Unlike typical 2-Cys PRDXs, the PRDX5 sulfinic form cannot be reduced by the sulfiredoxin system. Both Cys sulfinic and sulfonic acids are therefore irreversible. In vitro, DTT can be used to reduce the disulfide form, but not the hyperoxidized forms. To maintain the protein in a reduce state, the thiol can be alkylated by iodoacetamide (IAM) to prevent re-oxidation (r/aPRDX5, light green), or the protein can be point-mutated to replace the C47 by a serine (PRDX5^C47S^, dark green).

**Figure 2 antioxidants-10-01902-f002:**
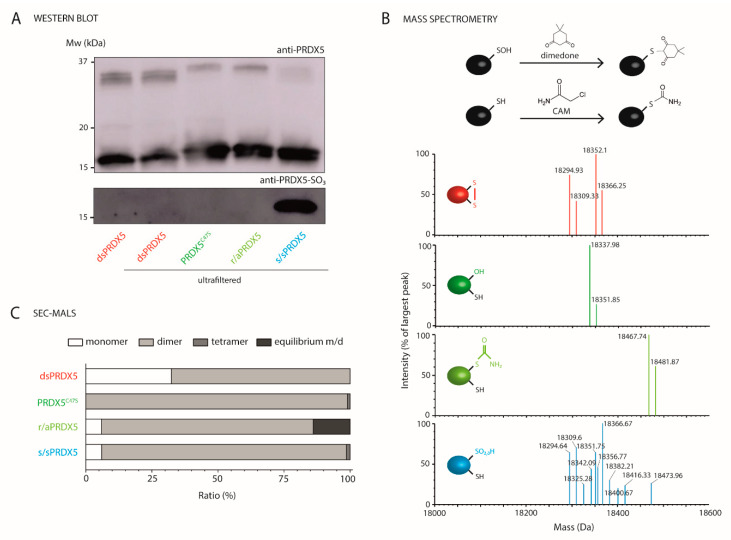
Characterization of the PRDX5 redox forms. (**A**) Western blot analysis of PRDX5 redox forms performed under non-reducing conditions. Western blotting against PRDX5 reveals the presence of monomers and dimers (upper) and presence of sulfonic acids was confirmed by Western blotting against PRDX5-SO_3_ and was only detected in s/sPRDX5 sample. dsPRDX5 was probed before and after ultrafiltration. Representative of three independent experiments. (**B**) Centroid spectrum showing monomeric species of the different redox forms (dimeric species are shown in Appendix A), generated after MaxEnt1 deconvolution of the multiply-charged spectrum obtained by direct infusion ESI-Q-TOF MS analysis on intact proteins. An intensity threshold of 20% normalized to the largest peak was applied to display the major peaks. dsPRDX5, r/aPRDX5, and s/sPRDX5 contains three Cys, while PRDX5^C47S^ has two remaining Cys. Prior to MS analysis, potent Cys sulfenic acids were trapped with dimedone, and remaining Cys thiols were blocked with 2-chloroacetamide (CAM). Identification of each peak is presented in Appendix A. (**C**) Quantification of the monomeric and oligomeric states using SEC-MALS. Source data are provided in Appendix A (blots), in Appendix A (MS multiply-charged spectra), and in Appendix A (SEC-MALS spectra).

**Figure 3 antioxidants-10-01902-f003:**
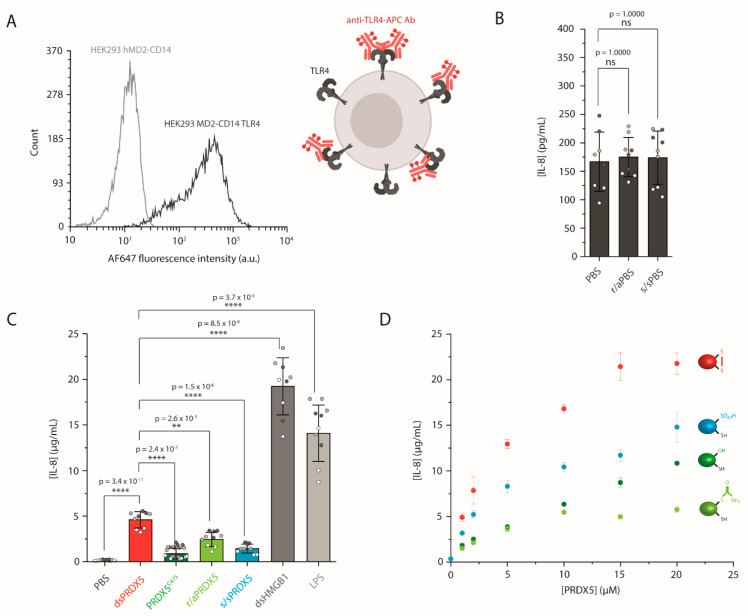
PRDX5 redox-dependent IL-8 secretion from HEK293 MD2-CD14 TLR4 cells. (**A**) Stable transfection of TLR4 in HEK293 MD2-CD14 cells was validated by flow cytometry (FC). Schematic depicting the labeling strategy used to validate the correct TLR4 expression at the cell surface. (**B**) Control experiment showing IL-8 production levels from HEK293 MD2-CD14 TLR4 cells treated with ultra-filtered PBS, “reduced/alkylated” PBS (r/aPBS) or “hyperoxidized” PBS (s/sPBS), quantified by ELISA. Three independent experiments (white, light grey and dark gray points) with three replicates per condition for each experiment. (**C**) IL-8 secretion from HEK293 MD2-CD14 TLR4 cells induced after stimulation with the PRDX5 redox forms, quantified by ELISA. LPS was used as a control to validate that TLR4 was functional, and HMGB1 was used as a DAMP control. Three independent experiments (white, light grey and dark gray points) with three replicates per condition for each experiment and two independent PRDX5^C47S^ protein productions. (**D**) Concentration-dependence of IL-8 secretion from HEK293 MD2-CD14 TLR4 cells for the redox forms. Data are representative of three replicates. Data are mean ± S.D. ns, not significant (*p* > 0.05). ** *p* ≤ 0.01, **** *p* ≤ 0.0001 (Tukey HSD test).

**Figure 4 antioxidants-10-01902-f004:**
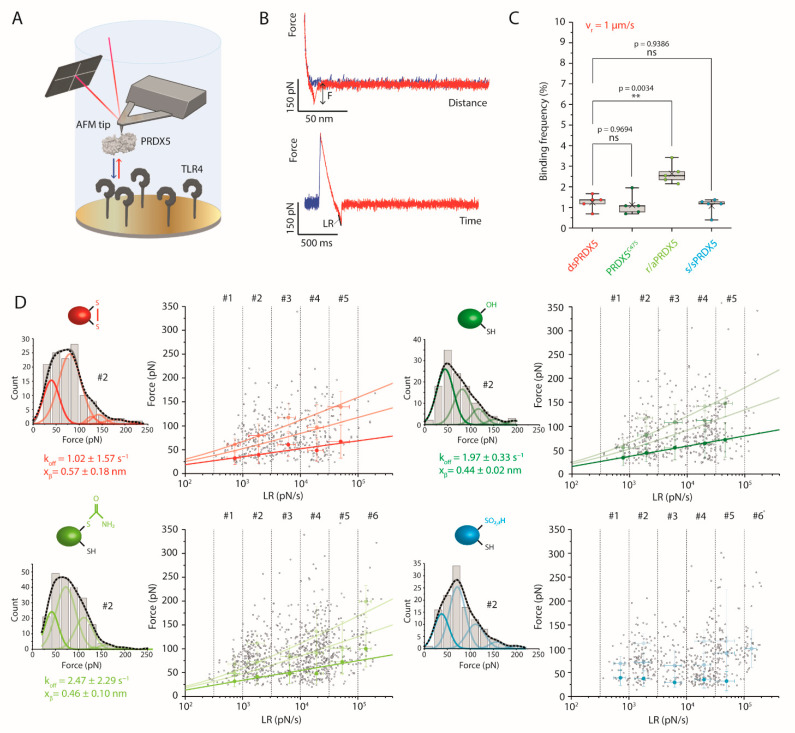
PRDX5 redox forms bind surface-immobilized TLR4 with similar $k_{off}$. (**A**) Cartoon showing the experimental set-up used to probe the interaction between the PRDX5 redox forms and TLR4. (**B**) Representative specific force-distance (FD) and force-time curves. Examples for all redox forms are shown in Appendix A. Rupture force (F) corresponds to the height of the adhesive peak on the FD curve, and the loading rate (LR) is calculated by fitting the adhesive peak on the force-time curve just before rupture with a linear model. (**C**) Box plot showing the binding frequencies (BF) obtained at a tip retraction speed of 1 µm·s^−1^. Number of AFM maps, *n* = 5 for each redox form (dots). Box plot shows mean (×), median (–), 25th and 75th percentiles (boxes), and ranges from min to max (whiskers). ns, not significant (*p* > 0.05). ** *p* ≤ 0.01 (Tukey HSD test). (**D**) Dynamic force spectroscopy (DFS) analysis for the PRDX5 redox forms. Spectrum of LRs was divided into discrete ranges (LR#1–6). For each LR range, a Kernel distribution of the rupture forces (grey line) was generated, as shown on the histogram for LR#2, which allows to determine hidden peaks based on the second derivative. Peaks were fitted with Gaussian functions (colored lines), revealing the most probable rupture forces for single (1st peak) and multiple (following peaks) interactions. These forces were reported on the DFS plot (full colored data points). The Bell-Evans model was fitted to data corresponding to single interactions (solid line), and the predicted rupture forces for two and three simultaneous uncorrelated interactions were calculated using Williams-Evans prediction (dashed lines). From the Bell-Evans model, the $k_{off}$ and $x_{\beta }$ values were extracted. The most probable rupture forces (full colored data points) are peak center ± S.D.

**Figure 5 antioxidants-10-01902-f005:**
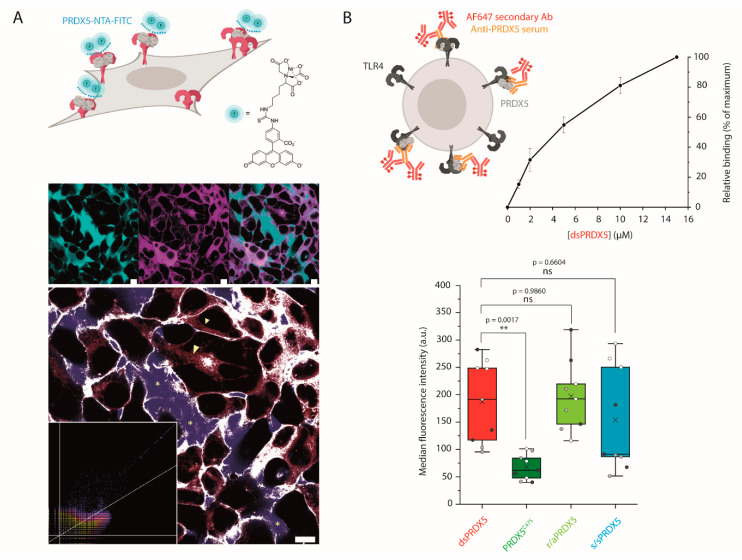
Binding of PRDX5 to the surface of HEK293 MD2-CD14 TLR4 cells monitored by fluorescence. (**A**) Cartoon (upper) showing the protocol used to detect PRDX5 binding to TLR4 using confocal. Confocal images of, left to right, PRDX5-NTA-FITC (cyan), TLR4-YFP (magenta), fluorescence overlay. Co-localization (white) of FITC (blue) and YFP (red) signals predominantly at the plasma membrane and in the endomembranous region. There is a lack of co-localization in the extracellular medium (*) and in YFP exclusive regions within cells (arrowhead). Co-localization map and scatterplot (inset). The cross-talk is represented by unique perfectly co-localized pixels. Scale bars are 10 µm. (**B**) Cartoon (upper) showing the protocol used to detect PRDX5 binding to living cells using flow cytometry (FC). Graph from FC binding assay showing the concentration dependence of dsPRDX5 binding to the surface of HEK293 MD2-CD14 TLR4 cells. Binding was normalized to the signal of untreated cells and is expressed relatively (in %) to the fluorescence intensity observed with 15 µM PRDX5. Three independent experiments with two replicates in each. Box blot (lower) from FC binding assay showing the median fluorescence intensity of PRDX5-treated cells. The fluorescence intensity was normalized to the untreated cells. Seven experiments with three independent protein treatments (white, light grey and dark grey points) or two protein productions in the case of PRDX5^C47S^. Three replicates for each redox form in each treatment. Box plot shows mean (×), median (–), 25th and 75th percentiles (boxes), and ranges from min to max (whiskers). ns, not significant (*p* > 0.05). ** *p* ≤ 0.01 (Tukey HSD test).

## Data Availability

Data is contained within the article and Appendix A.

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
