# Peer review of "Role of the Redox State of Human Peroxiredoxin-5 on Its TLR4-Activating DAMP Function"

_antioxidants, 2021, doi:10.3390/antiox10121902_

Round 1

Reviewer 1 Report

The authors have largely responded to the concerns raised from previous reviewers; in particular they undertook ESI-Q-TOF MS of intact proteins to characterize the modified forms of PRDX5 that they have produced for testing with TLR4-bearing cells.  I do not completely agree with conclusions but there is certainly more clarity with respect to the protein forms being used.  There is a lot of complexity to the MS results that bring up more questions in my mind (and it is curious that chloroacetamide has been used, although it gives the same product as iodoacetamide so the rationale is unclear), but for the most part the reader will have information that helps them understand the forms. 

Concern 1.  After going in detail through all the MS data and interpretations, I think at a minimum (as requested previously) that the authors must at least mention the third cysteine that is present, either in Fig. 2 legend or in the text that describes the results, since otherwise it makes no sense to end up with the 3 alkylated residues of the “r,a” form, in particular.

Concern 2. I feel that calling the dsPRXD5 preparation “homogenous” (e.g. on line 417) is a little disingenuous.  Already in Fig. 2A that is clearly not the case (with upper bands that that authors are somehow not interpreting as covalent linkages between dimers, even though they undergo reduction).  If they would like to call it “largely homogeneous” then I could accept that. 

Concern 3.  The authors talk about the “k off” not being statistically different, in the manuscript and in the abstract.  This statement is true but the numbers that were obtained are so variable as to be rather less clear than what is implied.  The authors then conclude that TLR4 cannot discriminate between the forms, but what about “k on” and “Kd”, which appear not to be determined by this method?  Perhaps the authors want to soften their statement on this point.

Additional comments.

I have comments back to the authors about the His tag and the use of NEM, but these do not require changes.  I am surprised that it appears that only the His-tagged version of Prdx5 was ever studied in the literature, and I do not see from the description that true “wild type” was ever studied and compared.  Since this His-tagged protein has a great deal of activity with peroxynitrite etc, hopefully it has produced data that is more or less trustworthy, but this is still very surprising. 

I can see from a new figure (Fig. R3) that the upper (dimer) band of the disulfide containing protein is still there whether or not NEM is added, so probably it is not due to a thiol-disulfide exchange reaction occurring under denaturing conditions; but the readers are not privy to this information for some reason.  Leaving the NEM out is generally not the correct way to do prepare samples for this type of gel.

Reviewer 2 Report

The authors addressed all points raised in a satisfactory manner.

Reviewer 3 Report

The authors have done an outstanding job at addressing the reviewers' questions with new data and editing figures and text. I do not have additional concerns. 

Author Response

This manuscript is a resubmission of an earlier submission. The following is a list of the peer review reports and author responses from that submission.

Round 1

Reviewer 1 Report

This manuscript by Poncin et al. reports experiments conducted with recombinant PRDX5, or a mutant,  treated in various ways, then applied to cells or studied in other ways, to investigate its linkage to TLR4 signaling and production of inflammatory cytokines, specifically IL-8.  With the initial experiments (Fig. 2), the authors offer data characterizing the varying redox and oligomeric states of their PRDX5 preparations, then conduct experiments adding each to HEK 293 (MD2-CD14 TLR4) cells to elicit IL-8 production, or tethering the PRDX5 proteins in AFM and conducting binding studies.  I do not profess much expertise in the cell work or AFM-related studies, but I do have expertise in PRDX protein function and analyses.  My major concern throughout is the limited knowledge the investigators have of the redox status of the PRDX5 mixtures they have created, the extreme heterogeneity of the samples (which could be avoided, at least in part), and particularly the 6xHis tag which remains on the protein throughout and is likely not insignificant.  It is (increasingly) well known in the field that such tags, and especially His tags, even at the N-terminus, considerably alter the oligomeric and functional state of these proteins.  In addition to prior data demonstrating such problems, a recent study has shown that N-terminally His tagged PRDX2 has very little activity.  It is unclear for this project that any prior work has validated, through full enzymological and biophysical characterization, that the His-tagged version of the wild type protein is unaltered.  Thus I have little enthusiasm for the rest of the studies at the outset that rely on this modified form of the protein (at least cleavage to remove the His tag would be advisable).  The heterogeneity and inadequate characterization of the various “redox forms” then compounds the problem.  In my view, the conclusions are not supported by the approaches used and data provided.  I am providing additional comments to try to help advise where problems may be arising with the various approaches.

The characterization of the obviously heterogenous samples intended to represent distinct redox states is based on modified peptides detected by LC-MS (which even so indicate much heterogeneity); showing these three replicates (?) as concentric circles is a highly problematic way to present these data.  Without standards for each type of modified peptide there can be huge differences in the ionization efficiency of each of those, skewing the estimated distribution.  Best would be to use electrospray MS to evaluate the mixtures of the INTACT proteins (where the modification barely changes the ionization properties of the full length protein).

Throughout, there is no mention of the additional Cys residue in this protein that should also be taken into account with the varying treatments (particularly when denaturing gels are being done).  Multiple technical aspects need attention in this work.  For example, any SDS gels (both reducing or non-reducing – note that the NR gel of Fig. 2a is not described as such in the legend, only in the text) should be run with samples alkylated (typically with NEM) before and during denaturation in the sample buffer.  Without that, Cys redox states can change, and particularly thiol-disulfide interchange reactions can change the locations of the disulfide bonds in the proteins.  This seems very evident in the covalent dimers that are observed in Fig. 2C, with little comment (even though they show in Fig. 1 that the native disulfide is formed within the monomer).

The very large standard deviations in the numbers for “IL-8 stimulating activity” as reported in the text (lines 466 to 475), at greater than 1/3 of the value in several cases, is in part a reflection of the poor degree of purity of each of the forms.  The authors should certainly not be giving so many (in)significant figures for each of these.  One would also expect to see the dose response curves for each of these in Fig. 3D, not just the disulfide form. (It would in fact behoove the authors to stabilize an appropriately intrasubunit-linked disulfide form, once generated, by alkylating it – on C72 that is not mentioned and any other free cysteines in the sample – so that the various disulfide “shuffled” forms, e.g. with C47 linked to C47 of another monomer, cannot form.)

To prepare hyperoxidized protein, a single addition, even at 5 mM, of H2O2 to prereduced protein will not be sufficient at all.  Even with the more sensitive PRDX proteins, one would make sure to promote cycling of the protein, including DTT in the mixture, or making alternate additions of H2O2 and DTT.  Once the protein is in disulfide form, it becomes quite insensitive to H2O2, except perhaps at the third Cys (not the one being studied/emphasized), thus multiple cycles are needed to allow progressively more protein to become hyperoxidized.

“Reduced” protein is alkylated, so it should not be called “reduced”.  Maybe “non-disulfide” if that is the point, or “reduced, alkylate”.  If in fact there is only 60% alkylation at C47, then the alkylation time should be extended, with additional DTT if needed to push the reaction to completion.  Note that the alkylation will occur (as long as accessible) on all three cysteine residues, not just the one (shown in Fig. 1B, which is misleading).

Minor

It is unclear where the antibody against hyperoxidized PRDX5 comes from; the reference given is to a paper that indicates what peptide could be used to elicit such an antibody, but did the authors actually obtain antibody from the Woo or Rhee labs?  To say that the Western blot analyzes “efficiency” of the hyperoxidation is certainly an overstatement.

It is incorrect to say that a sulfenic acid is “further oxidized” to a disulfide (line 48); in fact that is formally a reduction (from sulfenic acid state to disulfide, 0 to -1) while the other partner, a thiol, goes from -2 to -1, an oxidation.  It is also incorrect to state that sulfenic acid is “unstable” (and especially then reference a paper reporting a structure, 1prx, where the sulfenic acid is in fact very stable).  Perhaps the word “reactive” would be a better choice, or even “metastable”.

The authors are apparently unaware that iodoacetamide reacts with sulfenic acid as well, probably more slowly than with the thiol, and that it would give a product 16 mass units higher than the carboxamidomethyl product.

The pdb codes for the structures shown in Fig. 1 should be included in the legend.

In my view the “covalently grafted” PRDX5 on the AFM tip, as well as the TLR4 preparation, are not very physiologically informative, and getting only the “k off” from these experiments – which noticeably didn’t vary much – does not really provide much information.  Why not use surface plasmon resonance (even though I would also be suspicious of data from such an approach if not rigorously evaluated).

Reviewer 2 Report

The manuscript by Poncin et al. is an intersting piece of work concerning the role of Prdx5. The study is well carried out and results are clearly exposed.

I have only some minor points to address to the attention of the authors.

1) Since Prdxs are antioxidant proteins the kinetic parameters for the reported recombinant species should be measured and correlated to the oligomeric forms detected under the various tested conditions (activity versus Fig2C oligomeric sizes). Is enzymatic activity related to the biological activities examined by the authors on TLR4-dependent IL-8 secretion and on Prdx5-TLR4 binding kinetics?

2) As for Fig. 1A the reported crystal structures were from which pdb coordinates? how did the authors obtain the different shown conformational changes? are these already deposited structures?

Reviewer 3 Report

The manuscript describes an interesting study on the function of PRDX5 redox state in the activation of TLR4 during inflammation. While the manuscript has strengths, there are also significant deficiencies outlined below. A potential major concern is the calculation of the % redox modifications in the various forms of recombinant protein, which can mislead the interpretation of the SEC-MALS and other results. 

  1. line 53, delete “typical”
  2. Fig 2A, please specify in the figure legend that this is non-reducing blot. Is the -SO2/3 blot also nonreducing? Looks like yes, please specify this in the fig legend. It would be beneficial to also analyze the proteins by reducing blot to determine if the upper bands are stabilized by intermolecular disulfides.
  3. 2B scheme, IAM will likely react with both thiols, please correct this.
  4. lines 400-416 Has the difference in the ionization efficiency between carbamidomethylated, dimedone adduct, -SO3H, and disulfide peptides been considered when calculating the % of redox states?
  5. The AFM kinetic model only accounts for non-covalent binding. A mechanism could be envisioned where the disulfide in the dsPRDX5 is transferred to TLR4 (reaction with TLR4 thiol leading to intermolecular PRDX5-TLR4 disulfide which is then resolved by another thiol in TLR4 releasing rPRDX5 or internalization). A time course experiment could be performed to investigate this mechanism. Also to further establish the selectivity for PRDX5, the TLR4 experiments should include dsPRDX2 or dsPRDX4 as controls.
